# Why 'one size fits all' is not enough when designing COVID-19 immunity certificates for domestic use: a UK-wide cross-sectional online survey

Corina Elena Niculaescu [1], Isabel Sassoon [1], Irma Cecilia Landa-Avila [2], Ozlem Colak [2], Gyuchan Thomas Jun [2], Panagiotis Balatsoukas [2]

¹Department of Computer Science, Brunel University London, Uxbridge, UK
²School of Design and Creative Arts, Loughborough University, Loughborough, UK

**Correspondence to**
Dr Isabel Sassoon;
isabel.sassoon@brunel.ac.uk

## ABSTRACT

**Objectives** The present study explored public's willingness to use COVID-19 immunity certificates across six different domestic scenarios.

**Design** Cross-sectional online survey.

**Setting** UK representative survey conducted on 3 August 2021.

**Participants** 534 UK residents over 18 years old.

**Interventions** Participants replied to the same set of questions.

**Primary and secondary outcome measures** The primary outcome measure was *willingness to use* immunity certificates across three different domestic settings: (1) visiting the general practitioner (GP) for a non-urgent health issue; (2) dining in a restaurant and (3) attending a performance in a theatre. For each setting two options, one prioritising *convenience* (option A) and the other *privacy* (option B), were offered. Our secondary outcome measures were computed indices from items adapted from the Health Belief Model; attitudes towards sharing immunity status with service providers; prior to COVID-19 lifestyle. In addition, we recorded data about respondents' sociodemographic characteristics.

**Results** Respondents were more willing to use immunity certificates that prioritised *convenience (92%)*, rather than *privacy (76%)*, when visiting their GP . However, privacy was more favourable in the other two settings (dining in a restaurant (84%) and going to a theatre (83%)) compared with *convenience* (38% and 39% respectively). Personal beliefs about COVID-19 and immunity certificates were associated with variations in willingness to use these across all scenarios. No variations were observed across sociodemographics and lifestyle.

**Conclusions** The findings of this survey suggest that there is not *one-size-fits-all* solution for designing immunity certificates. Immunity certificates are complex sociotechnical systems, any attempt to implement these for domestic use should be tailored to different settings and user needs. The design of certification services requires a more evidence-based approach and further research is needed to understand how different settings, design elements (like *convenience* or *privacy*) and personal beliefs about the pandemic should inform their design.

## Strengths and limitations of this study

► The study reports knowledge about the interaction between individual characteristics, domestic settings and types of immunity certificate design on willingness to use these certificates.

► UK nationally representative sample for age, gender and ethnic background, but limited to people who have the means and capacity to use digital technologies (survey administered using Prolific.co).

► Since, as to the writing of this paper, COVID-19 certification has not been mandated in the UK, the scenarios used in the survey were hypothetical.

► We employed a Generalised Linear Mixed Effects Model to analyse whether there was a significant difference in the likelihood of using option A (*convenience*) and B (*privacy*) in each of the three settings.

## INTRODUCTION

Since the beginning of the COVID-19 pandemic immunity or vaccine certificates and their adoption by the public for domestic use has sparked a debate. The source for this debate can be attributed to several factors such as the uncertainty around the concept of immunity itself (eg, what does it really mean to be immune against COVID-19 and how long does this last?),[1–3] the almost antagonist tension between the protection of public health and the respect for human rights or civil liberties (eg, potential for creating inequality between those who are 'immunoprivileged' and those who are 'immunodeprived') as well as loss of autonomy,[2 4–8] legal challenges in implementing COVID-19 certification across different industries,[9 10] risk of fraud[11] and identity theft[5] and fear of digital exclusion.[12–14] Nonetheless, alongside the aforementioned concerns some potential benefits of immunity certificates have been reported such as preserving freedom of movement,[15] re-opening the economy,[16] reducing

BMJ

risk of infection and social benefits from increased social engagement.[12 16]

The debate on the above issues has been reported both among scientists and the public. In a survey of 12 738 scientists from 63 countries, while roughly 22% of the respondents reported concerns around inequality, more than half of them agreed that immunity certificates would be beneficial for the economy and public health.[16] In a different study, textual analysis of Twitter sentiment also showed that in the UK and the USA, vaccine certificates were associated with positive points among the public such as economic recovery, return to normality, safety, return to work or international travel, alongside negative connotations of discrimination,[14] surveillance or civil liberties.[17] Similar were the findings reported by another UK nationwide questionnaire survey exploring public attitudes towards vaccine passports.[18] The findings of this survey also showed that people from ethnic minority backgrounds and lower income (<£20 000) would feel unfairly discriminated from the use of vaccine passports.

In the present study, we approach immunity certification as a complex sociotechnical system. Unlike previous published research, we hypothesise that in order to understand what is the best way to design immunity certificates for domestic use we need to investigate the role of different contextual and situational factors, including different types of designs for COVID-19 certification, settings, individual characteristics and their interaction.

The present paper aims to progress the conversation around COVID-19 certification by answering two main research questions. First, would a design that promotes *convenience* or *privacy* increase willingness to use immunity certificates across three different domestic settings ((1) visiting the general practitioner (GP) for a non-urgent health issue, (2) dining in a restaurant and (3) attending a performance in a theatre)? Second, what is the role of sociodemographic characteristics, lifestyle, attitudes towards service providers and personal beliefs about COVID-19 on the aforementioned question? To address these two questions, we ran a nationally representative online questionnaire survey in the UK. Our findings produced unique knowledge about the interaction between individual characteristics, settings of use and types of certificate design on willingness to use immunity certificates. This knowledge can inform current policy on COVID-19 certification and lead to further development of existing solutions for domestic use, such as the UK's National Health Service (NHS) COVID-19 Pass.[19]

In the present paper, the term 'immunity certificate' refers to a type of certification (digital or paper) showing that an individual has developed antibodies of SARS-COV-2 either through past infection or after completing a full course of vaccination. We chose to focus on the concepts of *convenience* and *privacy* because both concepts have been reported frequently in the literature as key factors of user experience influencing use and/or adoption of these certificates among the public.[13 20–24] Purposefully, we selected to present *convenience* and *privacy* as two

extreme options to understand how willingness to use immunity certificates may be affected across different settings and individual characteristics. This decision was influenced by the findings of our previous studies where research participants perceived the two concepts not as complementary components to the design of services for immunity certificates but as antagonist elements that bring tension and dilemmas.[25] Therefore, in the context of the present study, the *privacy* option involved an individual installing the NHS app, accessing the COVID-19 certificate, generating or downloading a two-dimensional (2D) barcode and presenting this to the service provider for validation. In this option, the user of the service does not share any personal data electronically and the service provider, for example, the restaurant only scans the 2D barcode and manually checks the details in the certificate against the individual's form of identification. On the other hand, we hypothesised *convenience* as a situation where the individual would not need to download or instal an app and generate or download and share 2D barcodes. In the case of this scenario, we hypothesised that it would be more convenient for individuals to share their NHS number with the service provider. Then the service provider would use this number to verify someone's immunity status directly with the NHS (for instance, by checking it against the records held in the National Immunisation Management System). For example, when visiting a theatre to watch a performance the theatre company will verify a customer's immunity status directly with the NHS using the customer's NHS number. In both cases, it is the responsibility of the service provider to validate an individual's immunity status but in the case of the *privacy* option the customer or service user needs to go through a process that requires more physical and cognitive effort, while in the case of the convenience option the individual only shares their NHS number (without the need to instal any apps or generate and share barcodes). In the case of the convenience option, the individual is required to share personal information (ie, NHS number) with the service provider, while in the case of the privacy option, the service provider only validates the generated or downloaded barcode without digitally processing personal information (like the NHS number).[26]

Finally, we selected to focus on the specific three domestic settings (GP appointment for a non-urgent health matter, dining in a restaurant and going to a theatre) because all three represent common, yet distinct (in terms of purpose and social behaviours evoked), types of services that take place indoors.

## METHODS

### Design and methods

Our analysis was based on a cross-sectional dataset generated from an online questionnaire survey that took place on the 3 August 2021. The online questionnaire was created using the *onlinesurveys.ac.uk* platform and administered via *prolific.co*. All the materials related to this survey

including survey instrument, raw dataset, statistical code and ethics approval are available on Open Science Framework (OSF) (https://osf.io/jubv6/ DOI:10.17605/OSF. IO/JUBV6).

## Sample design

Respondents were demographically representative of the UK population in terms of gender, age and ethnicity. Summary statistics for all demographic variables can be found in the online supplemental material. We excluded 20 participants who failed the attention checks, and one duplicate responder, resulting in a final sample of 534 respondents. All participants were 18 years or older and were compensated for their participation in the study with £1.75/person.

## Patient and public involvement

The research questions and contents of the survey were informed by the findings of a series of studies (composed of focus groups and interviews), which involved a diverse group of research participants including patients, members of the public and service providers from the cultural, sports, hospitality and aviation sectors.[25]

## Main variables measure

### Willingness to use immunity certificates across different scenarios (primary outcome)

Each scenario was presented in the form of a short narrative description of a hypothetical use case that combined one of the three settings of interest, as described in the 'Introduction' section: (1) visiting the GP for a non-urgent health issue; (2) dining in a restaurant and (3) attending a performance in a theatre, with one of the following two options: (A) convenience and (B) privacy. Since the process of using COVID-19 certification for domestic purposes has not been mandated to date in the UK, and there is lot of speculation about how these could become operational in practice, the six scenarios explored hypothetical or future situations balancing imagination/creativity with rigorous reasoning techniques.[25 27] For illustration purposes, figure 1 presents the six scenarios. In the case of the *convenience* option, the service providers in each setting (eg, restaurant management or theatre company) had the authority to check a customer's COVID-19 immunity status with the NHS, without any additional steps on behalf of the users to prove their status. In the other option (*privacy option*), according to the scenario users needed to generate and share a QR code on booking an appointment or making a reservation with the service provider to demonstrate their immunity status. In this case, users were actively in control of the process of sharing their immunity status but they had to perform a series of actions to generate/ obtain and share the QR code.[20 25]

In each of the six scenarios, willingness to use immunity certificates was measured by asking respondents to rate how likely they were to use these based on a 5-point Likert scale (ranging from 'extremely unlikely' to 'extremely likely'). Following the data collection, positive to neutral answers, including 'extremely likely', 'somewhat likely' and 'neither likely nor unlikely' were grouped together under 'yes' describing willingness to use the service. Negative answers including 'extremely unlikely' and 'somewhat unlikely' were grouped together under 'no' describing reluctance to use the service. We dichotomised the scale this way to make its interpretation easier, and to differentiate between those willing to use the service (neutral to positive answers) and those resistant (negative answers). A similar transformation of 5-point Likert scale likelihood variables was used in previously published research reporting results from a series of surveys on adherence to test, trace and isolation measures in the UK.[28]

### Health Belief Model (secondary outcome)

We used a selection of items adapted from the Health Belief Model (HBM)[29] to examine whether certain health beliefs towards vaccination and COVID-19 could influence respondents' willingness to use immunity certificates across the six scenarios. The detailed description of the items, summary statistics and internal reliability measures are presented in table 1. Each item was rated on a 5-point Likert scale from 1 ('strongly disagree') to 5 ('strongly agree'). First, we measured *COVID-19 susceptibility* (respondent's perceived susceptibility) using three items adapted from Coe *et al*[30] and one item from Chu and Liu.[31] Second, we measured *certificate severity* (as the perceived severity of not using immunity certificates) using an index of six items. Additional HBM constructs were measured in our survey, but their analysis is beyond the scope of this paper and therefore these are not shown in table 1.

As it is shown in table 1, *COVID-19 susceptibility* generated a Cronbach's α of 0.7095 and certificate severity a score of 0.8485 suggesting good internal consistency.[32] Therefore, we created an index (*certificate severity* and *COVID-19 susceptibility*) for each of these two constructs by averaging the items within the constructs.[20] Then we used these indices to explore whether there were potential factors influencing willingness to use immunity certificates across the different scenarios.

### General attitudes towards sharing immunity status with service providers (secondary outcome)

Respondents were asked also to rate their willingness to share their immunity status with the following types of service providers: primary care GP/dentist, airport/ airline, hospitality (pub/restaurant and hotel), cultural and creative industries (theatres/cinemas/galleries) and sports event organisers (such as football clubs). Responses were recorded using a 5-point Likert scale ranging from 1 ('strongly disagree') to 5 ('strongly agree'). We used these items to examine whether willingness to use immunity certificates measured by our primary outcome was affected by willingness to share immunity status with different service providers in general.

**Scenario 1 (Visiting the GP for a non-urgent health matter)**
You want to book a face-to-face appointment with your GP for a non-urgent matter. A non-urgent matter is one that does not warrant using 111 or 999 NHS services. In order to ensure patient and medical personnel safety, your GP practice has health and safety measures in place requiring patients to prove their COVID-19 (SARS-CoV-2) immunity status when booking a face-to-face appointment. Proof of your immunity status will determine the location and time of the appointment. Now, let us consider two different ways in which this service could be implemented:

> **Option 1A, Convenience:** your GP practice has the authority to check your immunity status upon booking the appointment by checking the information in your NHS patient electronic health record (you will have to share your NHS number). There are no additional steps on your behalf.

> **Option 1B, Privacy:** your GP practice is not authorised to check your immunity status by accessing your NHS electronic health record. Instead, you would need either extract a QR code from your smartphone's NHS app and share it with your GP practice before confirming your booking or presenting a formal paper certificate proving your status.

**Scenario 2 (Dining in a restaurant)**
You want to go out for a meal at a restaurant, which only has indoor tables. In order to ensure the safety of the staff and the other diners, the restaurant has health and safety measures in place requiring customers to prove their COVID-19 (SARS-CoV-2) immunity status when making a reservation. Proof of your immunity status will enable you to dine at this restaurant. If you choose not to share your status, you would not be able to dine at this restaurant or book a table. Now, let us consider two different ways in which this service could be implemented:

> **Option 2A, Convenience:** the restaurant has the authority to check your status with the NHS upon booking a table or at least 48 hours prior. You would share your NHS number with the restaurant to enable them to check your immunity status. You would then be able to dine at the restaurant at your booked time without any extra steps on your behalf.

> **Option 2B, Privacy:** the restaurant has no authority to check your immunity status with the NHS on your behalf. Instead, you would need either to extract a QR code from your smartphone's NHS app and share it with the restaurant or present a formal paper certificate upon confirming your reservation.

**Scenario 3 (Attending a performance in the theatre)**
You want to attend a play at your local theatre, this is an indoor event. In order to ensure the safety of the audience, actors and the other staff, the theatre has health and safety measures in place requiring customers to prove their COVID-19 (SARS-CoV-2) immunity status when booking a ticket for a play. Proof of your immunity status will enable you to attend this play. If you choose not to share your status, you would not be able to attend this indoor event on the selected day. Now, let us consider two different ways in which this service could be implemented:

> **Option 3A, Convenience:** the theatre company has the authority to check your status with the NHS upon booking a ticket to a play or at least 48 hours prior. You would share your NHS number with the theatre company to enable them to check your immunity status. You would then be able to attend the play without any extra steps on your behalf.

> **Option 3B, Privacy:** the theatre company has no authority to check your immunity status with the NHS on your behalf. Instead, you would need either to extract a QR code from your smartphone's NHS app and share it with the theatre company or present a formal paper certificate upon confirming your booking or on the day of the play.

**Figure 1** Description of the six scenarios (the number represents one of the three settings while the letter the design option, convenience or privacy). GP, general practitioner; NHS, National Health Service.

### Prior to COVID-19 lifestyle (secondary outcome)

We asked a series of lifestyle questions to determine if respondents' habits before the COVID-19 outbreak correlated with the willingness to use immunity certificates in the different scenarios. Lifestyle questions recorded the respondents' perceived frequency of attending or

**Table 1** Summary statistics and reliability of HBM measures, and willingness to share immunity status

|  | Survey items | Mean | Median | SD | Minimum | Maximum | Alpha |
|---|---|---|---|---|---|---|---|
| COVID-19 perceived susceptibility (HBM) | I am at risk of getting COVID-19 (SARS-CoV-2) | 3.5243 | 4 | 1.1255 | 1 | 5 | 0.7095 |
|  | It is likely that I will get COVID-19 (SARS-COV-2) | 2.9401 | 3 | 1.0122 | 1 | 5 |  |
|  | Individuals in my household are at risk for getting COVID-19 (SARS-COV-2) | 3.4438 | 4 | 1.1310 | 1 | 5 |  |
|  | I feel knowledgeable about my risk of getting COVID-19 (SARS-COV-2) | 4.1255 | 4 | 0.7460 | 1 | 5 |  |
| Certificate severity (HBM) | I feel that without this service I will not be able to return to my workplace | 2.4476 | 2 | 1.1558 | 1 | 5 | 0.8485 |
|  | I feel that without this service my chances of getting a job will be affected | 2.5918 | 3 | 1.1631 | 1 | 5 |  |
|  | I feel that without this service I will not be able to book face-to-face appointments with my GP/dentist | 2.8371 | 3 | 1.2455 | 1 | 5 |  |
|  | I feel that without this service I will not be able to go to the theatre/movies/sports events | 3.2715 | 4 | 1.1636 | 1 | 5 |  |
|  | I feel that without this service I will not be able to travel internationally | 3.912 | 4 | 1.1252 | 1 | 5 |  |
|  | I feel that without this service I will not enjoy the same liberties I did before the pandemic | 3.6667 | 4 | 1.1692 | 1 | 5 |  |
| Willingness to share immunity status with service providers | Theatre/Cinema/Gallery | 3.2921 | 4 | 1.3998 | 1 | 5 | – |
|  | Pub/Restaurant | 3.2228 | 4 | 1.4159 | 1 | 5 |  |
|  | GP/Dentist | 4.47 | 5 | 0.9219 | 1 | 5 |  |
|  | Hospitality sector | 3.4663 | 4 | 1.3717 | 1 | 5 |  |
|  | Sports event | 3.3015 | 4 | 1.4012 | 1 | 5 |  |
|  | Airport/Airline | 3.8764 | 4 | 1.2538 | 1 | 5 |  |

GP, general practitioner; HBM, Health Belief Model.

pursuing various activities of immediate interest to the scenarios under investigation including going to the theatre or other cultural venues (like museums and galleries), going to pubs, restaurants and other dining venues or visiting healthcare settings. Other lifestyle questions collected data about their frequency of travelling abroad, and booking accommodation when travelling abroad. Responses were captured using a 4-point Likert scale ranging from 1 ('never') to 4 ('very often').

### Statistical analysis

To address the first research question, this survey design enabled us to analyse the responses collected for the primary outcome measure (ie, willingness to use immunity certificates across different scenarios) as a repeated measures 2×3 factorial design with two fixed effects (the *convenience/privacy options* and the setting) and with a random effect of the responder. The dependent variable for this analysis was the willingness to use immunity certificates. Each responder had six willingness to use responses each corresponding to a different scenario, which was the result of the combination of *convenience* versus *privacy* options and settings.[33]

To analyse whether there was a significant difference in the likelihood of using option A (*convenience*) and B (*privacy*) in each of the three settings, we employed a Generalised Linear Mixed Effects Model (GLMM).[33 34] We fit the GLMM model which incorporates both fixed effects parameters (*convenience/privacy* and setting) and random effects in a linear predictor, via maximum likelihood.

Finally, to address our second research question, we employed an exploratory analysis through graphical

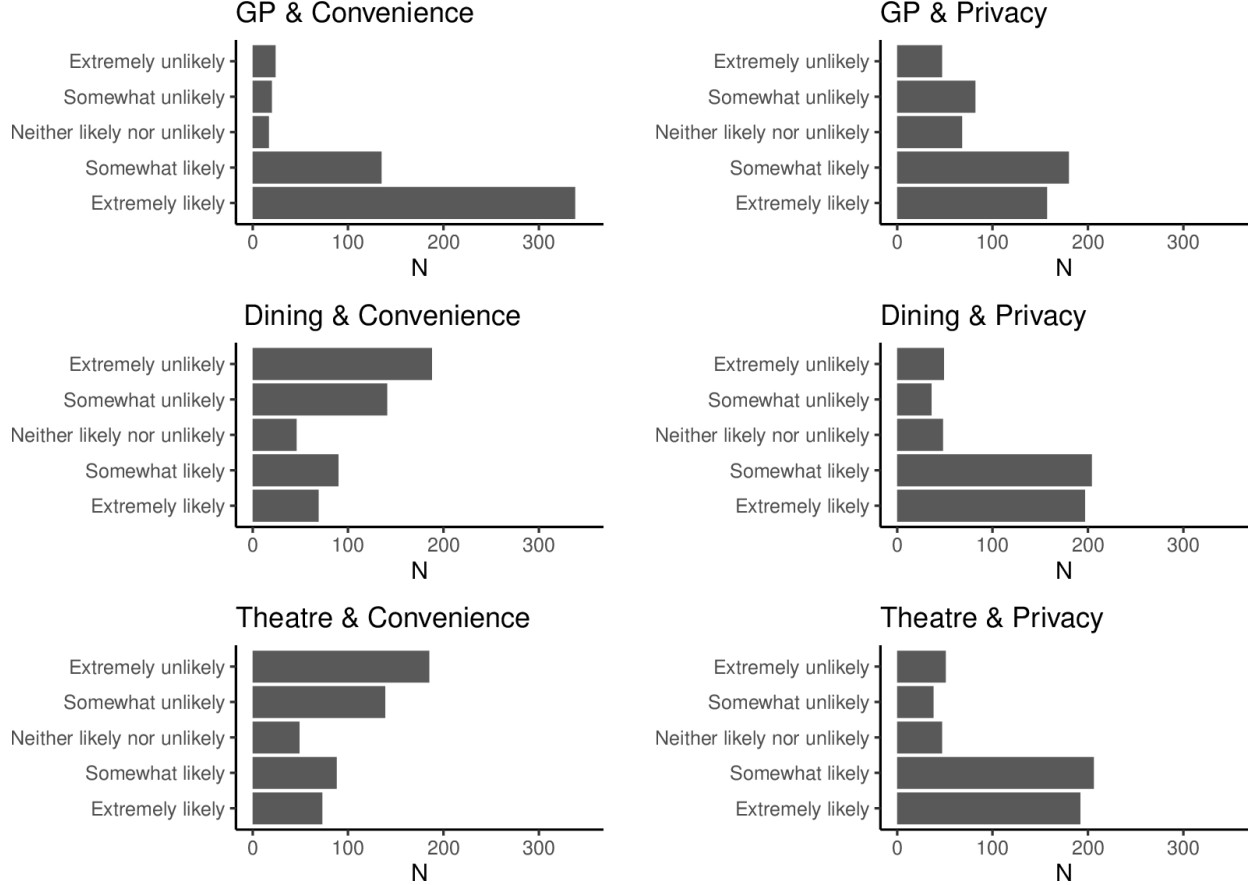

**Figure 2** Distribution of number of responses (N) across settings (visiting the general practitioner (GP), dining in a restaurant, attending a performance in the theatre) and design options (convenience/privacy).

representations of the relationships between the dependent variables (primary outcome measure) and the other secondary outcome measures. To display relationships between the primary outcome measure and the HBM indices we used box plot graphs, and for the rest of the variables we present mosaic plots. The statistical analysis was performed in STATA[35] and R.[36]

### Power calculation

The sample size was chosen pragmatically based on several different approaches, obtaining a minimum sample size between 271 and 1067 participants, depending on the assumptions. This sample size results in a 99% power in the GLMM model used in our statistical analysis of the first research question.

### RESULTS

### Does a person's willingness to use immunity certificates vary across the six scenarios?

Figure 2 illustrates the distribution of responses across the six scenarios while table 2 the proportion of respondents who would be willing to use the certificates across these scenarios. The data show that the majority of respondents (92%) were more willing to use immunity certificates that prioritised *convenience* when visiting their GP (scenario 1A). However, *convenience* was less favourable in the other

two settings with only 38% and 39% of respondents willing to use the certificates for dinning indoors (scenario 2A) and going to the theatre (scenario 3A), respectively.

To determine whether these differences in willingness to use were statistically significant, we applied the GLMM model, including fixed and random effects outlined in 'Statistical analysis' section. Table 3 includes a subset of

| Table 2 | Acceptance of different service designs and settings | | |
|---|---|---|
| **Response** | **No** | **Yes** |
| Scenario 1A: visiting the GP/convenience option | 44 | 490 (92%) |
| Scenario 1B: visiting the GP/privacy option | 129 | 405 (76%) |
| Scenario 2A: dining in a restaurant/convenience option | 329 | 205 (38%) |
| Scenario 2B: dining in a restaurant/privacy option | 85 | 449 (84%) |
| Scenario 3A: attending a performance in the theatre/convenience option | 324 | 210 (39%) |
| Scenario 3B: attending a performance in the theatre/privacy option | 89 | 445 (83%) |

**Table 3** Summary of the coefficients of the generalised linear mixed model fit

| Fixed effects | Estimate | SE | Z value | Pr (>|z|) |
|---|---|---|---|---|
| (Intercept) | 4.1636 | 0.2646 | 15.7300 | <2e-16*** |
| Dining in a restaurant setting | −5.0121 | 0.2837 | −17.6700 | <2e-16*** |
| Attending a performance in the theatre setting | −4.9398 | 0.2822 | −17.5000 | <2e-16*** |
| Privacy option | −2.0736 | 0.2504 | −8.2800 | <2e-16*** |
| Dining in a restaurant setting Setting* privacy option | 5.8747 | 0.3595 | 16.3400 | <2e-16*** |
| Attending a performance in the theatre Setting* privacy option | 5.7127 | 0.3557 | 16.0600 | <2e-16*** |

The coefficients in log-odds form of the GLMM model and their significance. The estimates of the fixed effects are conditional on the random effects. The estimated effects have a binary outcome with a logit link; hence, the raw estimates are on the log-odds scale. The intercept refers to the log-odds for willingness to use immunity certificates in scenario 1A (visiting the GP/convenience option). A positive value for log-odds estimates respondents being likely to be willing to use the certificate in that setting and option. The coefficients on the log-odds scale are additive.
Significance codes 0 '***'; 0.001 '**'; 0.01 '*'; 0.05 '.'; 0.1 ' '.
GLMM, Generalised Linear Mixed Effects Model; GP, general practitioner.

the R output for the GLMM analysis (logistic regression with mixed fixed and random effects). The estimated coefficient for the intercept 4.1636 is the log-odds for scenario 1A (choosing *convenience* option for visiting the GP for a non-urgent health issue). The estimate of −5.0121 for dinning in a restaurant means that this setting is associated with a 5.0121 decrease in the log-odds of positive response ('yes') compared with negative response ('no'). The log-odds for scenario 2B (choosing *privacy* for dining in a restaurant) is the sum of the intercept (4.1636), the indoor dining setting (−5.0121), the *privacy* option (−2.0736) and their interaction (5.8747), resulting in a log-odds of 2.9526. This points to a higher likelihood of willingness to use immunity certificates for scenario 2B (choosing *privacy* for dining in a restaurant) than for scenario 2A (choosing *convenience* for dining in a restaurant).

The likelihood of using the certificate when going a restaurant or to the theatre was significantly lower than visiting the GP with a statistically significant p <0.001 when considering the scenarios irrespective of the option. Also, the likelihood of using the certificate was significantly lower for *privacy* than *convenience* when visiting the GP (p<0.001). However, when considering the *privacy option* in the restaurant or theatre setting, this likelihood of using the certificate is higher and statistically significant (p<0.001).

### What is the role of personal beliefs about COVID-19 (certificate severity and COVID-19 susceptibility), attitudes towards service providers, lifestyle and sociodemographic characteristics on the willingness to use immunity certificates?

To address our second research question, we graphically explored the primary outcome variables against all other secondary outcome variables, but for brevity we only present here the key results.

► *Certificate severity.* Lower values for *certificate severity* suggest that respondents did not perceive immunity certificates as necessary (figure 3A). We observed lower perceived *certificate severity* among those who were not willing to use immunity certificates. This finding was observed across all settings and for both options (*convenience* and *privacy*).

► *COVID-19 susceptibility.* The median of *perceived susceptibility* is consistently higher for responses indicating willingness to use immunity certificates across all scenarios even when these prioritised *convenience* (figure 3B). This means that respondents who perceived themselves as being more susceptible to COVID-19 were more willing to use immunity certificates for both options. Also, the median values show that this group of respondents would be more willing to use an option that prioritised *convenience* when going to a restaurant or the theatre rather than when visiting their GP for a non-urgent health matter. This finding suggests that people who perceived themselves to be at high risk of COVID-19 were more willing to trade-off *privacy* for *convenience* for specific settings.

► *General attitudes towards sharing immunity status with service providers.* Respondents who were comfortable sharing their COVID-19 immunity status with service providers from the cultural and creative industries (eg, theatres, cinemas or galleries) were more willing to use immunity certificates across all three settings (figure 4). Also, this group of respondents was more likely to choose options prioritising *convenience*. On the contrary, respondents who did not like sharing their immunity status with service providers from the creative and cultural industries were less willing to use immunity certificates across all scenarios even when this prioritised *privacy*. Similar patterns with those exhibited in the case of cultural and creative industries were observed in the case of the other types of

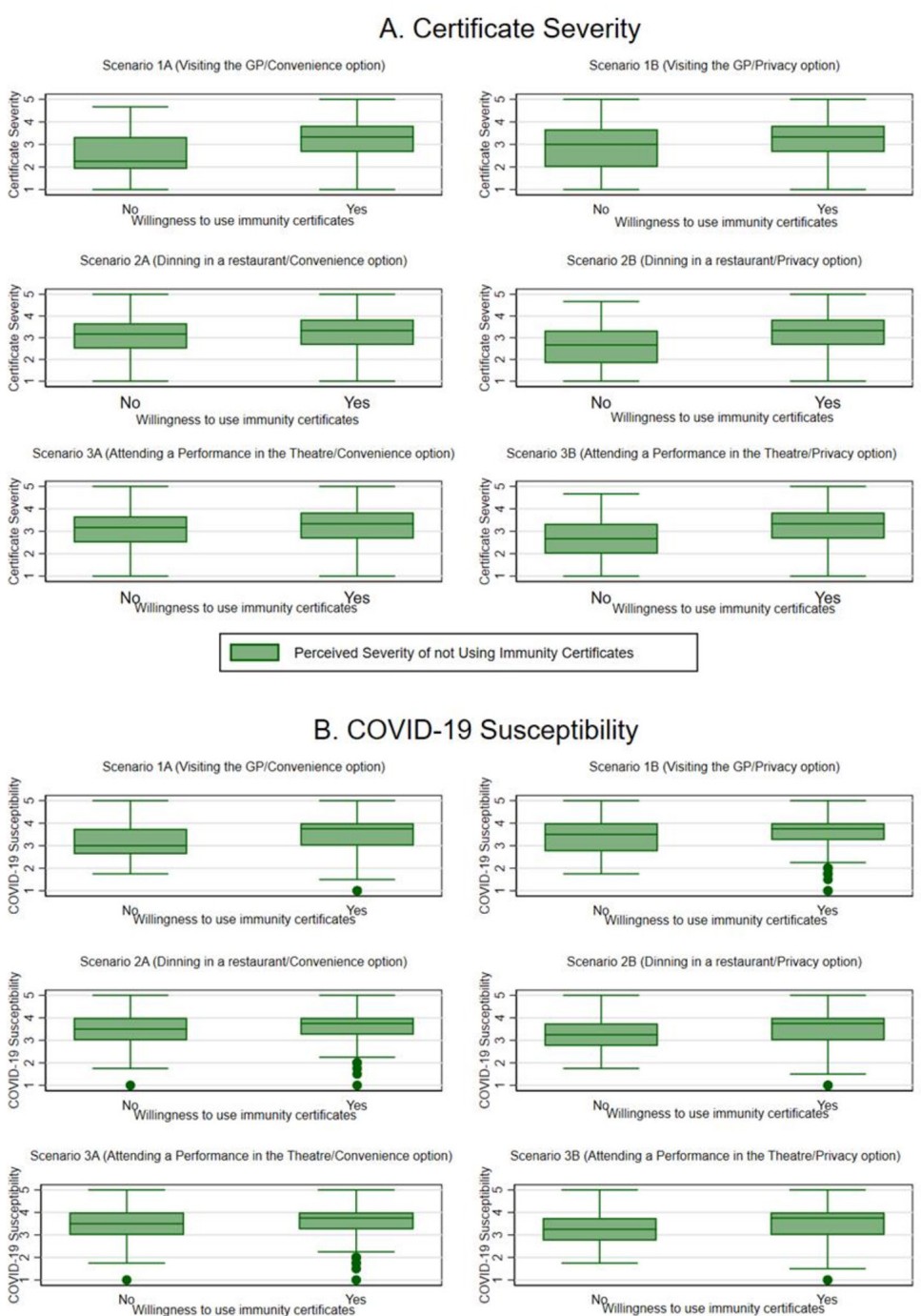

**Figure 3** (A) Index of certificate severity (perceived severity of not using immunity certificates) and (B) index of COVID-19 susceptibility across settings and design options (convenience/privacy).

service providers, including hospitality, airlines/airports, sports events and pubs/restaurants.

► *The role of sociodemographics and lifestyle characteristics.* Our exploration of the relationship between willingness to use the immunity certificates across the six scenarios did not point to any significant differences across sociodemographics or lifestyle. Specifically, we observed no variation in willingness to use immunity certificates across the different scenarios by gender, age and ethnicity. Also, willingness to use immunity certificates did not vary by other sociodemographics such as disability, living arrangements and living in a rural versus urban area. In addition, lifestyle characteristics were not associated with variations in willingness to use immunity certificates either. Lastly, there was no variation in willingness to use the service based on mental well-being and net income now as compared with before the outbreak, suggesting that willingness to use immunity certificates does not stream from feelings of hopelessness.

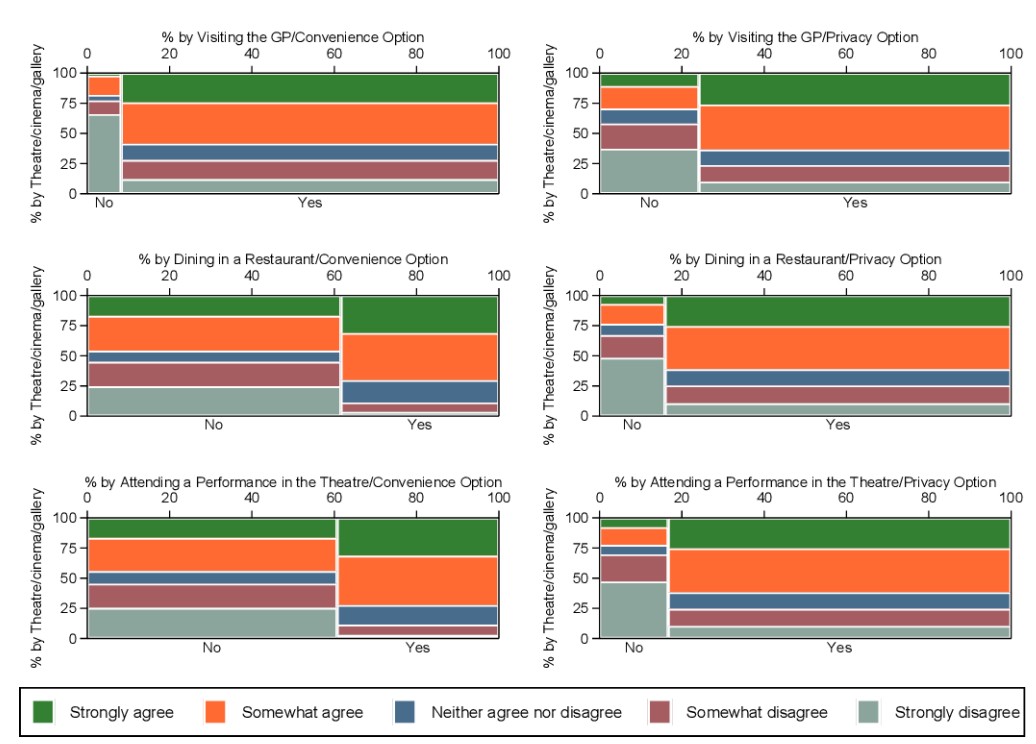

**Figure 4** Willingness to share immunity status with service providers (theatre/cinema/gallery) by across settings and design options (convenience/privacy).

## DISCUSSION

### Practical implication for policy making

As of the writing of this paper (September 2021), the use of immunity certificates for domestic purposes in the UK was not mandatory. Also, there was still lack of knowledge or guidance about what is the best way to design immunity certificates for use in different domestic settings. Therefore, the present survey contributes unique knowledge that different stakeholders (the NHS, the UK government and the Scientific Advisory Group for Emergencies, businesses and the public) should take into account when considering the use of immunity certificates for domestic purposes. Some key implications of this study for policy making are the following:

► *Use of certification in primary care settings.* Participants' responses showed high willingness to use immunity certificates when visiting their GP for a non-urgent health matter compared with the other settings (dinning in a restaurant and going to the theatre). In this case, the majority of responses were positive for both options (*privacy* and *convenience*). We argue that the implementation of immunity certificates for non-urgent healthcare matters could play a role in increasing the sense of safety as well as reducing waiting times for face-to-face appointments especially when the option that prioritises *convenience* is selected to confirm patients' immunity status.

However, as of today, in the UK, the use of any type of immunity certificate in healthcare settings is not recommended. Currently, consultations with a GP

can be either remote (videoconferencing or via telephone) or face-to-face. The transition to prepandemic appointments is not fully completed, and the risks of attending face-to-face appointments are still present. For instance, dental procedures can increase the risk of SARS-CoV-2 transmission due to tools that produce aerosols,[37 38] not to mention the issue of contracting the virus in a hospital setting.[39 40] As such, patients and healthcare professionals are presented with the choice between remote appointments for non-urgent matters or the risk of infection in a face-to-face setting. While to a certain extend there is some evidence that remote consultations have been accepted well by the patients,[41] several challenges have been reported too, especially in the case of people who suffer from pre-existing chronic conditions or who they may not feel comfortable with the use of technology.[41–43] For those patients, implementation of immunity certificates for face-to-face appointments would be important in improving their sense of safety. In addition, our findings showed that as opposed to the other domestic settings, in the case of visiting a GP for a non-urgent matter, respondents were more willing to use immunity certificates that prioritised *convenience*. The reason for choosing this option could be justified by the fact that respondents valued the public health benefits of using certificates in this context as more important compared with the option than prioritised privacy.[44] Also, this means that respondents in this survey preferred access to primary care services to be

seamless without a need for patients to show proof of their immunity status in a digital or physical format. Instead, this proof could be verified by the GP practice on booking an appointment (over the phone or electronically). This would result in zero checks at the reception, shorter queues on arrival and better experience for the patients.

► *Use of certification for social indoor activities.* When it comes to leisure activities (such as dining in a restaurant or going to a theatre), respondents were less willing to use immunity certificates, and if they did most of them would value the *privacy* option over *convenience.* One key implication of this finding when designing certification services is to make clear to the public how data *privacy* and protection applies to this context. The findings of our previous research[25] suggest that the public is not convinced about whether or not their data are shared when using these certificates, thus making them more sceptical to use these in settings where they feel that they are less secure or they do not trust. This happens despite the fact that in the UK verification of someone's immunity status using the paper or digital certification format (via the NHS app[19]) does not involve sharing of any personal information with the service provider (eg, the theatre company or the restaurant management team can only verify the validity of the barcode itself).[45] For the use and uptake of such certificates to be successful the public and businesses should be educated, and nationwide public health campaigns should promote this shared understanding explaining the extent to which personal data, if any, is shared on verification of someone's immunity status. Researchers examining the implementation of the European Union (EU)'s COVID-19 digital certificate for international travel (known as green pass) highlighted the importance of campaigns to educate the public about the purpose, opportunities and limitations of these certificates too.[44] In addition to educating the public and promoting the importance of certification for public health, the presence of such nationwide campaigns is important for transparency reasons too. Different countries around the globe have used immunity certificates in ways that made possible the collection and processing of large amounts of personal data about individuals for reasons other than merely proving their immunity status, for example, in order to monitor the flow of people in shops or other settings.[46] Explaining and communicating decisions and actions around public health to the public in a transparent way has been identified as a key leadership characteristic of public health authorities that was lacking during the first wave of this pandemic both in the EU[47] and the UK.[48]

► *Need to build empathy and understand public's views about COVID-19 when designing certification services.* While lots of attention when designing different forms of COVID-19 certification was placed on issues surrounding their accessibility,[14 49–51] less effort has been put in place to harness public's beliefs around COVID-19 and COVID-19 certification and fed these into the design of services for immunity certificates. The implementation of immunity certificates should be accompanied by a series of health promotion strategies tailored to target the needs of people with different beliefs, knowledge and understanding about COVID-19, and ultimately change their behaviour. The present survey showed that traditional demographic information and lifestyle does not influence user willingness to use immunity certificates across the six scenarios. However, our findings also showed that perceived risk of falling severely ill from COVID-19 (*COVID-19 susceptibility*) and perceived severity from not using immunity certificates (*certificate severity*) can influence public's willingness to use immunity certificates.

## Limitations

One of the limitations of our study is that participants were recruited using the online survey platform Prolific. co. Since surveys administered via this platform are completed online (mobile, PC, tablet, etc), our sample comprised people who had the means and capacity to use digital technologies. Other studies investigating people's perceptions of immunity certificates[14 20 52] or COVID-19 vaccine intentions[28 53–55] found some differences based on gender and ethnicity, which we did not find. This can be explained by the fact that unlike the other cited studies, our survey was focused on six specific hypothetical scenarios of using immunity certificates. Another possible explanation can be attributed to the timing of this survey. At the time our survey took place immunity certificates were being used for international travel, hence access, awareness and familiarity with such services were higher than in previously published studies. Finally, from a methodological point of view the mixed effects model we used assumed that the random effect subsumes the possible effects of gender and ethnicity, as we are looking at responses of the same person. Therefore, both the scope and timing of the present study as well as the mixed effects model used differed from other surveys in this context.

## CONCLUSIONS

The findings of this survey suggest that there is not *one-size-fits-all* solution for designing immunity certificates. Immunity certificates should be studied as complex sociotechnical services rather than merely products that one can simply download and use. Any attempt to implement such certificates for domestic use should be tailored to different settings and user needs. While some implications of our findings for policy making were discussed, the design of certification services requires a more evidence-based approach and further research is needed to examine willingness to use immunity certificates across the present three, and possibly other, domestic settings.

Also, while in the present survey we chose to explore two options for designing immunity certificates (one prioritising *convenience* and the other *privacy*), our previous work[56] has shown that there are still more attributes that may influence use in this context and future research should focus on.

**Contributors** The questionnaire survey was conceptualised by CEN, IS and PB, with the input of TG, ICL-A and OC. CEN and PB completed the data collection. CEN and IS conducted the statistical analysis. All authors contributed and approved the final manuscript. IS and PB are responsible for the overall content as guarantors.

**Funding** IMMUNE or Immunity Passport Service Design is a 9-month project funded by the AHRC/UKRI COVID-19 Rapid Response (Ref. AH/W000288/1).

**Competing interests** None declared.

**Patient and public involvement** Patients and/or the public were involved in the design, or conduct, or reporting, or dissemination plans of this research. Refer to the Methods section for further details.

**Patient consent for publication** Consent obtained directly from patient(s)

**Ethics approval** Ethics approval was obtained from the College of Engineering, Design and Physical Sciences Research Ethics Committee at Brunel University London (Ref. 31705-A-Jul/2021-33586-1) on 29 July 2021. Informed consent was obtained from all respondents prior to the beginning of the survey. Respondents were allowed to withdraw from the survey at any time.

**Provenance and peer review** Not commissioned; externally peer reviewed.

**Data availability statement** Data are available in a public, open access repository. Data are available in the project Open Science Framework (OSF) repository (https://osf.io/jubv6/).

**ORCID iDs**
Corina Elena Niculaescu http://orcid.org/0000-0002-8971-0812
Isabel Sassoon http://orcid.org/0000-0002-8685-1054
Irma Cecilia Landa-Avila http://orcid.org/0000-0001-6107-6736
Ozlem Colak http://orcid.org/0000-0003-0813-2561
Gyuchan Thomas Jun http://orcid.org/0000-0002-0958-0107
Panagiotis Balatsoukas http://orcid.org/0000-0001-9051-7633

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
