## [Reviewer comments · BMJ Open]

ARTICLE DETAILS

TITLE (PROVISIONAL)	Why "one size fits all" is not enough when designing COVID-19 immunity certificates for domestic use: a UK wide cross-sectional online survey
AUTHORS	Niculaescu, Corina; Sassoon, Isabel; Landa-Avila, Irma; Colak, Ozlem; Jun, Gyuchan; Balatsoukas, Panagiotis

VERSION 1 – REVIEW

REVIEWER	Hall, Mark Wake Forest University, School of Law
REVIEW RETURNED	08-Nov-2021

GENERAL COMMENTS	Well designed and well conducted and written study of an important public policy issue.
---

REVIEWER	Rocha, Ian Christopher Centro Escolar University
REVIEW RETURNED	14-Nov-2021

GENERAL COMMENTS	Thank you for submitting a very interesting manuscript. After thorough review, I am recommending minor revisions. In this regard, kindly address the following comments and suggestions to further improve your manuscript: 1) Your whole manuscript is well written, especially the introduction and discussion sections. I really commend the authors for choosing a very interesting topic to write on.2) You may also want to include the table of demographic characteristics of your participants in the main manuscript. I noticed that you submitted it as a supplementary material. However, I believe it is also important to include this table in the main manuscript for publication.3) Please double check the numbers/data in your results, especially the tables, to avoid erroneous findings of the study.4) In designing immunity certificates, you may want to read and cite this letter with related topic pertaining to using digital/online vaccination certificates in everyday life: https://academic.oup.com/jpubhealth/advance-article/doi/10.1093/pubmed/fdab308/6328303
--

REVIEWER	Marinelli, Enrico Sapienza University of Rome
REVIEW RETURNED	04-Dec-2021

GENERAL COMMENTS	I have read with interest the article titled Why “one size fits all” is not enough when designing COVID-19 immunity certificates for domestic use: a UK wide cross-sectional online survey”, which seeks to address the issue of willingness to use vaccination certificates on the part of the general public based on a survey gauging factors such as "convenience" and "privacy". The article is certainly original, fairly well written and even interesting in some parts, thanks to its solid methodology and thorough analysis of its stated objective. I am somewhat doubtful that it should be submitted to a medical journal, since there is nothing meaningful in terms of public health and pandemic management, and the authors have no medical background whatsoever. Still, the Editor will be the judge of that. I would say a few concepts need to be clarified: the notion of convenience is outlined as "the shift of responsibility for proving an individual’s immunity status, from the individual itself, to the service provider whose services the individual wishes to use. For example, when visiting a theatre to watch a performance it is the responsibility of the theatre company to verify a customer’s immunity status directly with the NHS without further involvement from the customer. Isn't that always the case? How can someone wishing to attend a play or dine at a restaurant be demanded to prove its status other than by a certificate linked to a database? On the other hand, privacy was defined as people being actively in control about sharing their immunity status and therefore being responsible to decide when, where and with whom their status could be shared. How can that be the case for someone seeking to gain access to a restaurant or movie theater, or board a plane? What would the "additional effort on their behalf be? Obtaining and sharing proof of their immunity status how? Doesn't that happen through linking to a database anyway? The European COVID-19 certificate certainly has all the proper features regarding privacy and convenience, and I believe the UK linked with the EU Gateway last 1 November, which enables mutual verification of COVID-19 certificates, so the logic at the root of that differentiation escapes me. It seem like the classical "distinction without a difference". In addition to clarifying those points, the authors ought to take into account different perspectives, outside the survey, in the Discussion. Maybe the people's willingness to use COVID passes has something to do with how these are perceived in terms of how effectively they can pursue public health objectives. In that regard, I would recommend drawing upon the following sources: Montanari Vergallo G, Zaami S, Negro F, Brunetti P, Del Rio A, Marinelli E. Does the EU COVID Digital Certificate Strike a Reasonable Balance between Mobility Needs and Public Health? Medicina (Kaunas). 2021 Oct 9;57(10):1077. Wang Q, Su M, Zhang M, Li R. Integrating Digital Technologies and Public Health to Fight Covid-19 Pandemic: Key Technologies, Applications, Challenges and Outlook of Digital Healthcare. Int J Environ Res Public Health. 2021 Jun 4;18(11):6053. Gontariuk M, Krafft T, Rehbock C, Townend D, Van der Auwermeulen L, Pilot E. The European Union and Public Health Emergencies: Expert Opinions on the Management of the First
--

	Wave of the COVID-19 Pandemic and Suggestions for Future Emergencies. Front Public Health. 2021 Aug 20;9:698995. Looking forward to reviewing a new improved version of this article.
--	--

VERSION 1 – AUTHOR RESPONSE

Reviewer	Comment -id	Reviewer comment	Author response
Editor	1	Revise the 'Strengths and limitations' section of your manuscript (after the abstract). This section should contain up to five short bullet points, no longer than one sentence each, that relate specifically to the methods. The novelty, aims, results or expected impact of the study should not be summarised here.	We have revised the phrasing of the "strength and limitations" as requested.
1	1	Well designed and well conducted and written study of an important public policy issue.	We thank the reviewer for the feedback.
2	1	Your whole manuscript is well written, especially the introduction and discussion sections. I really commend the authors for choosing a very interesting topic to write on.	We thank the reviewer for the feedback.
2	2	You may also want to include the table of demographic characteristics of your participants in the main manuscript. I noticed that you submitted it as a supplementary material. However, I believe it is also important to include this table in the main manuscript for publication.	We appreciate the suggestion, but we are working to a limit on tables and figures. If the editors agree to go over the limit, then we would be happy to include the table of demographic characteristics in the main manuscript.
2	3	Please double check the numbers/data in your results, especially the tables, to avoid erroneous findings of the study.	Thank you for pointing this out. We have double checked and made a correction in Section 2.2.2.

2	4	In designing immunity certificates, you may want to read and cite this letter with related topic pertaining to using digital/online vaccination certificates in everyday life: https://academic.oup.com/jpubhealth/advance-article/doi/10.1093/pubmed/fdab308/6328303	Thanks for pointing us to this pertinent resource. We have added this reference in the first paragraph of the Introduction section of the paper – reference number 11.
3	1	I would say a few concepts need to be clarified: the notion of convenience is outlined as "the shift of responsibility for proving an individual's immunity status, from the individual itself, to the service provider whose services the individual wishes to use. For example, when visiting a theatre to watch a performance it is the responsibility of the theatre company to verify a customer's immunity status directly with the NHS without further involvement from the customer. Isn't that always the case? How can someone wishing to attend a play or dine at a restaurant be demanded to prove its status other than by a certificate linked to a database? On the other hand, privacy was defined as people being actively in control about sharing their immunity status and therefore being responsible to decide when, where and with whom their status could be shared. How can that be the case for someone seeking to gain access to a restaurant or movie theater, or board a plane? What would the "additional effort on their behalf be? Obtaining and sharing proof of their immunity status how? Doesn't that happen through linking to a database anyway? The European COVID-19 certificate certainly has all the proper features regarding privacy and convenience, and I believe the UK linked with the EU	We are happy to provide further clarifications about the difference between the concepts of convenience and privacy. It is important to note that at the time this survey took place (3rd of August 2021) the UK government had not published any plans relating to the domestic use of immunity certificates. Immunity certificates were used for international travel but there was no legislation, guidance or evidence about whether these certificates were going to be used for domestic purposes, and how or where. This is also the reason why we named both options as hypothetical, because at the time of this study, and as opposed to other countries in Europe where immunity certificates were used domestically (like it happened, for example, in Italy or Greece) the whole of the UK did not follow the same policy. Also, later in December 2021 when immunity passes became mandatory in the UK for domestic use, this happened only for certain selected types of indoor venues only (in other countries, like Greece, for example, immunity passports became mandatory everywhere even for sitting in the open space/outdoor space of a café, or visiting the GP). The UK followed a completely different direction compared to other countries in Europe both when it comes to timings, venues and how these certificates were used domestically. Having explained why the two options are characterised as hypothetical (e.g. by the time of this survey there were no plans to use the certificates

		Gateway last 1 November, which enables mutual verification of COVID-19 certificates, so the logic at the root of that differentiation escapes me. It seem like the classical "distinction without a difference"	for domestic purposes), we can clarify now in more detail the definition of privacy and convenience in the context of the present study. (Below is the text that we have added to the introduction) “ Therefore, in the context of the present study, the privacy option involved an individual installing the NHS app, accessing the COVID-19 certificate, generating or downloading a 2D barcode and presenting this to the service provider for validation. In this option the user of the service does not share any personal data electronically and the service provider, for example the restaurant, only scans the 2D barcode and manually checks the details in the certificate against the individual’s form of identification. On the other hand, we hypothesised convenience as a situation where the individual would not need to download or install an app and generate or download and share 2D barcodes. In the case of this scenario we hypothesised that it would be more convenient for individuals to share their NHS number with the service provider. Then the service provider would use this number to verify someone's immunity status directly with the NHS (for instance, by checking it against the records held in the National Immunisation Management System NIMS). For example, when visiting a theatre to watch a performance the theatre company will verify a customer’s immunity status directly with the NHS using the customer’s NHS number. In both cases it is the responsibility of the service provider to validate an individual’s immunity status but in the case of the privacy option the customer or service user needs to go through a process that requires more physical and cognitive effort, while in the case of the
--	--	--	---

			convenience option the individual only shares their NHS number (without the need to install any apps or generate and share barcodes). In the case of the convenience option the individual is required to share personal information (i.e. NHS number) with the service provider, while in the case of the privacy option the service provider only validates the generated or downloaded barcode without digitally processing personal information (like the NHS number)[26]. “ We hope that the explanation presented above helps clarify the distinction between the privacy and convenience options. We revised the writing of the specific part of the manuscript (in the Introduction section) accordingly and we are confident that the revised version helps readers who might be less familiar with the UK context to comprehend the distinction between the two options clearly.
--	--	--	--

3	2	In addition to clarifying those points, the authors ought to take into account different perspectives, outside the survey, in the Discussion. Maybe the people's willingness to use COVID passes has something to do with how these are perceived in terms of how effectively they can pursue public health objectives. In that regard, I would recommend drawing upon the following sources: Montanari Vergallo G, Zaami S, Negro F, Brunetti P, Del Rio A, Marinelli E. Does the EU COVID Digital Certificate Strike a Reasonable Balance between Mobility Needs and Public Health? Medicina (Kaunas). 2021 Oct 9;57(10):1077. Wang Q, Su M, Zhang M, Li R. Integrating Digital Technologies and Public Health to Fight Covid-19 Pandemic: Key Technologies, Applications, Challenges and Outlook of Digital Healthcare. Int J Environ Res Public Health. 2021 Jun 4;18(11):6053. Gontariuk M, Krafft T, Rehbock C, Townend D, Van der Auwermeulen L, Pilot E. The European Union and Public Health Emergencies: Expert Opinions on the Management of the First Wave of the COVID-19 Pandemic and Suggestions for Future Emergencies. Front Public Health. 2021 Aug 20;9:698995.	Thank you for sharing these articles with us. We have incorporated these in the discussion section (section 5) as references 44, 46 and 47 respectively of our manuscript to further support our arguments.
---	---	---	--

VERSION 2 – REVIEW

REVIEWER	Rocha, Ian Christopher Centro Escolar University
REVIEW RETURNED	19-Mar-2022

GENERAL COMMENTS	Thank you very much for improving and revising your manuscript based on the comments and suggestions of the reviewers. After thorough review of your revised manuscript, I am hereby recommending acceptance of your manuscript for publication.
--

REVIEWER	Marinelli, Enrico Sapienza University of Rome
-----------------	--

REVIEW RETURNED	10-Mar-2022
-------------

GENERAL COMMENTS	I feel the points I have raised have been addressed quite adequately and thoroughly, more elaboration and contextualization have been provided and as a result, the article is in my estimation more well-rounded and comprehensive. An element of comparative discussion has been added and I believe the article should now be deemed worthy of publication, by virtue of its relevance, sound methodology and originality.
---